

ERROR APPORTIONMENT FOR ATMOSPHERIC CHEMISTRY-TRANSPORT
MODELS. A NEW APPROACH TO MODEL EVALUATION
E. Solazzo, S. Galmarini
European Commission, Joint Research Centre, Institute for Environment and Sustainability,
Air and Climate Unit, Ispra, Italy
Author for correspondence: S. Galmarini, stefano.galmarini@jrc.ec.europa.eu,
Phone: +390332785382
**Abstract.** In this study, methods are proposed to diagnose the causes of errors in air quality
(AQ) modelling systems. We investigate the deviation between modelled and observed time
series of surface ozone through a revised formulation for breaking down the mean square
error (MSE) into bias, variance, and the minimum achievable MSE (*mMSE*). The bias
measures the accuracy and implies the existence of systematic errors and poor
representation of data complexity, the variance measures the precision and provides an
estimate of the variability of the modelling results in relation to the observed data, and the
*mMSE* reflects unsystematic errors and provides a measure of the associativity between the
modelled and the observed fields through the correlation coefficient. Each of the error
components is analysed independently and apportioned to resolved process based on the
corresponding timescale (long scale, synoptic, diurnal, and intra-day) and as a function of
model complexity.
The apportionment of the error is applied to the AQMEII (Air Quality Model Evaluation
International Initiative) group of models, which embrace the majority of regional AQ
modelling systems currently used in Europe and North America.
The proposed technique has proven to be a compact estimator of the operational metrics
commonly used for model evaluation (bias, variance, and correlation coefficient), and has
the further benefit of apportioning the error to the originating timescale, thus allowing for a
clearer diagnosis of the process that caused the error.
*Keywords:* Model evaluation; Time series analysis; Bias-variance decomposition; AQMEII
1. INTRODUCTION
Due to their use for regulatory applications and to support legislation, air quality (AQ)
models must model correctly and be correctly applied, justifying the need for a thorough
evaluation. A framework for the operational and scientific evaluation of geophysical models
was already envisaged in the early '80s (Fox, 1981; Wilmott et al., 1985), the former being '*a*
*comparison with data exclusively within a particular application context*', and the latter
defined as '*some understanding of cause-and-effect relationship that relies on testing model*
*components and extensively detailed data collection*' (Fox, 1981). Thirty years later, as AQ



models became more and more complex and their range of applicability widened, Dennis et
al. (2010) further elaborated the concept of model evaluation by proposing a four-level
evaluation, according to which different complementary aspects of the models should be
tested, namely:
a. Operational: the level of agreement of model results with observations;
b. Dynamic: ability of the modelling system to respond to changes (in emissions, or in
meteorological events);
c. Diagnostic: identify and attribute the source of the error to the relevant process;
d. Probabilistic: confidence and uncertainty levels of the modelled results.
In the framework originally designed by Dennis et al. (2010), the diagnostic component
plays a central role. It *i)* answers the fundamental issue left open by the operational
screening, in other words whether the model provides the right answer for the right reason,
*ii)* provides feedback to developers to help make model improvements, and *iii)* sets the
basis for the probabilistic evaluation (Figure 1 of Dennis et al., 2010).
Over the years, and despite the increasing relevance of modelling systems for AQ
applications, model evaluation continues to rely almost exclusively on operational
evaluation, which basically involves gauging the model's performance using distance,
variability, and associativity metrics. This common practice has little or no impact on model
improvement, as it does not target the source of the modelling error and does not
discriminate between the reasons for appropriate or inappropriate performance.
Such a requirement is even more pressing these days, with current state-of-the-science AQ
modelling systems accounting for an increasing number of coupled physical processes and
being described using hundreds of modules, which are the result of decades of targeted
and, generally, independent investigations. Furthermore, AQ modelling systems typically
depend on external sources for the inputs of meteorology and emissions data, as well as for
boundary conditions. These fields are generally produced by other models (which, in turn,
depend on external sources for initial and/or boundary conditions) and, after substantial
processing, are used by the AQ modelling systems with no guarantee of being unbiased
and/or accurate. The bias introduced by these inputs, along with the uncertainty associated
with model error, the linearisation of non-linear processes, and omitted and unresolved
variables and processes, all contribute to the model error. The extensive use of AQ models
for AQ assessment and planning is equally important, and requires a good knowledge of the
model capabilities and deficiencies that would allow for a more educated use of the
modelling systems and their results.
Recently, the AQMEII (Air Quality Model Evaluation International Initiative) activity (Rao et
al., 2011) applied the approach proposed by Dennis et al. (2010), by organising model





evaluation activities (AQMEII 1, 2 and 3) using operational (Solazzo et al., 2012a,b; Solazzo
et al., 2013a; Im et al., 2015a,b), probabilistic (Solazzo et al., 2013b; Kioutsioukis et al.,
2014), and diagnostic (Hogrefe et al., 2014; Makar et al., 2015) evaluation frameworks.
The study we present here follows and complements the previous investigations based on
the AQMEII models collected in the first and second phases of the activity (AQMEII1 in 2006
and AQMEII2 in 2010). The main aim is to introduce a novel method that combines
operational and diagnostic evaluations. This method helps apportion the model error to its
components, thereby identifying the space/timescale at which it is most relevant and, when
possible, to infer which process/es could have generated it. This work is designed to support
the analysis of the currently ongoing third phase of the AQMEII activity (Galmarini et al.,
83  2015).

2. MEAN SQUARE ERROR AS A COMPREHENSIVE METRIC
For the model evaluation strategy proposed, we start by breaking down the Mean Square
Error (MSE) (used here as unique metric to evaluate model performance) into the sum of
the variance (and covariance) and the squared bias. The error and its components are then
calculated on the spectrally decomposed time series of modelled and observed hourly
ozone mixing ratios. The advantage of this evaluation strategy is twofold:
• With respect to a conventional operational evaluation, the new method allows for a
more detailed assessment of the distance between model results and observations
given the breakdown of the error into bias, variance and covariance and their
associated interpretations.
• Decomposing the MSE into spectral signals allows for the precise identification of
where each portion of the model error predominantly occurs. Given that specific
processes are associated with specific scales, the apportion of the error
components to their relevant scales helps to more precisely identify which processes
described in the model could be responsible for the error. Information about the
nature of the error and the class of process can significantly help modellers and
developers to improve model performance.
The data used are produced by the modelling communities participating in AQMEII1 and
AQMEII2 over the European (EU) and North American (NA) continental scale domains for
the years 2006 (AQMEII1) and 2010 (AQMEII2).
2. 1. ERROR DECOMPOSITION
The MSE is the squared difference of the modelled (*mod*) and observed (*obs*) values:

$$MSE = E(mod - obs)^2 = \frac{\sum_{i=1}^{nt}(mod_i - obs_i)^2}{n_t}$$

EQ 1

where $E(\cdot)$ denotes expectation and $n_t$ is the length of the time series. The bias is:



$$bias = E(mod - obs)$$ **EQ 2**

i.e. $bias = \overline{mod} - \overline{obs}$ . Thus, the following relationship holds:

$$MSE = var(mod - obs) + bias^2$$ **EQ 3**


which is a well-known property of the MSE, (var(·) is the variance operator). By using the
property of the variance for correlated fields:

$$var(mod - obs) = var(mod) + var(obs) - 2cov(mod, obs)$$ **EQ 4**


the final formulation for the MSE components reads:

$$MSE = bias^2 + var(mod) + var(obs) - 2cov(mod, obs),$$ **EQ 5**


where the covariance term (last term on the right-hand side of Eq 5) accounts for the
degree of correlation between the modelled and observed time series. When the covariance
term is zero, *var(obs)* is referred to as the *incompressible part of the error* and represents
the lowest limit that the MSE of the model can achieve. When dealing with model
evaluation, the modelled and observed time series are typically highly correlated and
therefore, within the limits of the perfect match (correlation coefficient of unity), *cov(mod,*
*obs) = cov(obs,obs) = cov(mod,mod) = var(mod) = var(obs)* and the MSE can be reduced to
only the bias term. That implies that the development of a high-quality model needs to
ensure:
*a.* the highest possible precision in order to maximise the *cov(mod, obs)* term, and
*b.* the highest possible accuracy, in order to minimise the bias.
Elaborating on Eq 5, Theil (1961) derived the following:

$$MSE = (\overline{mod} - \overline{obs})^2 + (\sigma_{mod} - \sigma_{obs})^2 + 2(1 - r)\sigma_{mod}\sigma_{obs}$$ **EQ 6**


In Eq 6, the variance term is expressed as the difference between the standard deviation of
the model and that of the observations, and the covariance term (last term on the right)
includes *r*, the coefficient of correlation between the observed and modelled time series.
The ratios of the three terms on the right-hand side of Eq 6 to the overall MSE are known as
*Theil's coefficients* (Pindick and Rubinfeld, 1998).
The bias measures the departure of the modelled from the observed results, and is a
measure of systematic error, since it measures the extent to which the average modelled
values deviate from the observed ones. The bias is commonly used to express the degree of
'trueness', i.e. "the closeness of agreement between the average value obtained from a
large series of measurements and the true value" (Johnson, 2008). The variance shows





whether the modelled variability is compatible with that observed. Finally, the covariance
term represents the unexplained proportion of the MSE due to the remaining unsystematic
errors, i.e. it represents the remaining error after deviations from the mean values have
been accounted for. This latter term is a measure of the lack of correlation of the model
with comparable observations, and is considered the least 'worrisome' portion of the error
(Pindick and Rubinfeld, 1998).
Elaborating on Eq 6, the conditions that minimise the MSE are:

$$\begin{cases} \dfrac{\partial MSE}{\partial \overline{mod}} = 2(\overline{mod} - \overline{obs}) = 0 \\ \dfrac{\partial MSE}{\partial \sigma_{mod}} = 2(\sigma_m - \sigma_{obs}) + 2(1-r)\sigma_{obs} = 0 \end{cases}$$

i.e. the best agreement between modelled and observed values is achieved by:

$$\begin{cases} \overline{mod} = \overline{obs} \\ \sigma_m = r\sigma_{obs} \end{cases} \qquad \text{EQ 7}$$


which analytically corresponds to the aforementioned items *a* and *b*. By inserting Eq 7 into
Eq 6, the minimum achievable MSE (*mMSE*) is

$$mMSE = \sigma_{obs}^2(1 - r^2) \qquad \text{EQ 8}$$


which is the unexplained portion of the error, as it reflects the share of observed variance
that is not explained by the model ($r^2$ is the coefficient of determination). The presence of
an unexplained part of the error suggests a modification of the MSE decomposition in Eq 6
in such a way as to explicitly include *mMSE*:

$$MSE = (\overline{mod} - \overline{obs})^2 + (\sigma_{mod} - r\sigma_{obs})^2 + mMSE \qquad \text{EQ 9}$$


The decompositions in Eq 5, Eq 6, and Eq 9 contain all the relevant operational metrics
usually applied to score modelling systems (bias, variance, correlation coefficient), and
therefore prove to be a compact estimator of accuracy (bias), precision (variance) and
associativity (unexplained portion through the correlation coefficient). Eq 9 has been
explicitly derived in this study to help evaluate AQ models. Murphy (1988) provided
examples of the scores that can be developed using the components of the MSE.
Ideally, the entire error should be attributable to unsystematic fluctuations. From a model
development perspective, the variance and covariance are possibly more revealing of model
deficiencies than is the bias term, as they are produced by the AQ model itself, while the
bias is also due to external sources (e.g. emissions, boundary conditions). From the





application viewpoint, however, it is the overall error that counts, which is mostly made up
of the bias.
2.2. SPECTRAL DECOMPOSITION OF MODELLED AND OBSERVED TIME SERIES
Hourly time series of (modelled and observed) ozone concentrations have been
decomposed using an iterative moving average approach known as the Kolmogorov-
Zurbenko (kz) low-pass filter (Zurbenko, 1986), whose applications to ozone are vastly
documented in the literature (Rao et al., 1997; Wise and Comrie, 2005; Hogrefe et al., 2000
and 2014; Galmarini et al., 2013; Kang et al., 2013; Solazzo and Galmarini, 2015). The kz
filter depends on two parameters: the length of the moving average window $m$ and the
number of iterations $k$ ($kz_{m,k}$). Since the kz is a low-pass filter, the filtered time series
consists of the low-frequency fluctuating component, while the difference between two
filtered time series provides a band-pass filter. This latter property is used to decompose the
ozone concentration time series as:

$$O_3 = LT(O_3) + SY(O_3) + DU(O_3) + ID(O_3)$$     EQ 10


where LT is the long-term component (periods longer than 21 days); SY is the synoptic
component (weather processes that last between 2.5 and 21 days); DU is the diurnal
component (day/night alternation period between 0.5 and 2.5 days); and ID is the intra-day
component accounting for fast-acting processes (less than 12 hours). The decomposition
presented in Eq 10 is such that the original time series is perfectly returned by the
summation of the components (see Appendix for details). Dealing with one year of data, any
filter longer than the LT component would not be meaningful. The periods of the
components correspond to well-defined peaks in the power spectrum of ozone, e.g. as
detailed in Rao et al. (1997) and Hogrefe et al. (2000).
The LT component is the baseline and incorporates the bias of the original (undecomposed)
time series. The other components (SY, DU, and ID) are zero-mean fluctuations around the
LT time series and are therefore unbiased. The band-pass nature of the SY, DU, and ID
components is such that they only account for the processes occurring in the time window
the filter allows the signal to 'pass'. For instance, the DU component is insensitive to
processes outside the range of 0.5 to 2.5 days.
Further properties of the spectrally decomposed ozone time series of AQMEII derived by
Galmarini et al. (2013), Hogrefe et al. (2014), and Solazzo and Galmarini (2015) are as
follows:
-    The DU component accounts for more than half of the total variance, followed by
the LT and SY components;

-    The ID component has the smallest influence due to the small amplitude of its
fluctuations;



- The variance of the spectral component is neither strongly nor systematically associated with the area-type of the monitoring stations (i.e. rural, urban, suburban);

- Due to the bias, most of the error is accounted for by the LT component, followed by the DU component. The ID contributes very little to the overall MSE.

Further important technicalities of the spectral decomposition, including a method to estimate the contribution of the spectral cross-components (the overlapping regions of the power spectrum) to the total error, are reported in the Appendix.

The signal decomposition of Eq 10 is applied to the full-year time series. However, to evaluate the model performance with regard to ozone, the analysis is restricted to the months of May to September, i.e. when the production of ozone due to photochemistry is most relevant.

## 3. DATA AND MODELS USED

The observational dataset derived from the surface AQ monitoring networks operating in the EU and NA constitutes the same dataset used in the first and second phases of AQMEII to support model evaluation. Only stations with over 75% valid records for the whole periods and located at altitudes below 1 000 m have been used for this analysis. Details of the modelled regions and number of receptor stations are reported in Table 1.

Since the main scope of this study is to introduce the error apportionment methodology (rather than to strictly evaluate the models), the analysis is presented for continental areas for convenience and easier display of the results. However, given the size of the domains and the heterogeneity of climatic and emission conditions, dedicated analyses for three sub-regions in both continents are proposed in the Supplementary material (Figure S1 to Figure S3).

There are profound differences between the modelling systems that participated in AQMEII1 and AQMEII2. The two sets of models have been applied to different years (2006 for phase 1 and 2010 for phase 2) and are therefore dissimilar with respect to the input data of emissions and boundary conditions for chemistry. The AQ models of the second phase are coupled (online chemistry feedbacks on meteorology), while those of the first phase are not. The effect of using online models for simulating ozone accounts for the impact of aerosols on radiation and therefore on temperature and photolysis rates (Baklanov et al., 2014).

The model settings and input data for phase I are described in Solazzo et al. (2012a, b; 2013a), Schere et al. (2012), and Pouliot et al. (2012); for phase II, similar information is presented in Im et al. (2015a, b), Brunner et al. (2015), and Pouliot et al. (2015).

Table 2 summarises the features of the modelling systems analysed in this study with regard to ozone concentrations in the EU or NA. The modelling contribution to the two phases of AQMEII consists of 12 and 9 models and of 8 and 3 models for EU and NA, respectively.





Detailed analysis of the main differences in emissions, boundary conditions, and
meteorology between the modelled years of 2006 (AQMEII1) and 2010 (AQMEII2) is
presented in Stoeckenius et al. (2015). A summary of the performance of the two suites of
model runs is provided in Makar et al. (2015), showing that the AQMEII1 models generally
performed better than the AQMEII2 models, based on standard operational metrics.
However, the use of standard evaluation methods does not allow for the assessment of
whether the feedback processes have an effect on the deterioration of model performance,
or rather the different sets of emissions and boundary conditions. We try to assess the
problem using the error apportionment methods outlined above.
## 4. RESULTS FOR THE SPATIALLY AVERAGED TIME SERIES
### 4.1 MSE OF SPECTRAL COMPONENTS
Figure 1 reports the MSE share of the spectral components and cross components for each
model, for both phases of AQMEII, spatially averaged over the two continental areas.
The LT share of the total MSE is the largest in absolute value for both continents and both
simulated years. The LT share ranges between 9.9% (GEM-AQ, AQMEII1, NA) and 86.7%
(WRF/Chem, AQMEII1, NA), and averages at ~34% and ~46.5% for the EU and ~50.6% and
~47% for NA (AQMEII1 and AQMEII2, respectively).
The second largest share of the total MSE is of the DU component, accounting for ~20% (all
cases), followed by the SY component. Depending on the model, the MSE share of the
remaining spectral components and cross-components varies significantly. Being the
intermediate time scales, the overlap of the DU and SY components is likely to be more
significant than the overlap of the LT and ID scales. The contribution of $DU_{cc}$ and $SY_{cc}$ to the
total error can be as high as 17% ($DU_{cc}$ for GEM-AQ, AQMEII1, NA) and 16% ($SY_{cc}$ for MM5-
CAMx, AQMEII1, EU). Overall, the $DU_{cc}$ terms (interaction of DU with the neighbouring SY
and ID scales) are significant in both continents (~10%), while the share of the SY
component and cross-components is more significant in the EU.
The ID component has a little impact or negligible on the total MSE (negligible in some
instances), exceeding the 3% share only for the two EU instances of the L.-Euros model.
The results of Figure 1 help identify the time-scales and associated processes for which the
largest improvement in model accuracy can be achieved. The LT component has the largest
share of the error due to the bias (error breakdown is discussed in the next section), but
'internal' chemical processes, transport, and deposition also occur at this timescale.  Diurnal
processes are the second largest source of error, including, among others, chemistry,
boundary layer dynamics, radiation forcing, and their interactions. The processes in the SY
band bridge meteorological and chemical processes, and discern between the fast-acting
diurnal processes and the baseline. As such, although the SY signal is not as strong as that of





the DU components (variance of SY is comparable to the variance of ID, see Hogrefe et al.,
2014), it accounts for a significant portion of the total error, as discussed next.

### 4.2 THE QUALITY OF THE ERROR: ERROR APPORTIONMENT

The error breakdown (Eq 9) of each spectral component complements the analysis
presented in the previous section, and is reported in Figure 2. The bias (only included in the LT
component) is the average amount by which the modelled time series is displaced with
respect to the observed time series, and is the main source of error. The bias can be either
due to 'internal' model errors, or inherited from external drivers (emissions, meteorology,
boundary conditions). While the former are of interest for model development because
they are generated by systematic modelling errors, the bias introduced by external drivers is
responsible for the largest share of modelling errors.
From the continental average error breakdown of Figure 2 we can conclude that the majority
of EU models (in both AQMEII phases) have small bias (continental-wide average), with the
important exceptions of CCLM-CMAQ and Muscat models in AQMEII1, and CMAQ in
AQMEII2, which introduced large positive biases. The bias for the NA continent is more
uniformly distributed across the models (model over-prediction in both AQMEII phases),
possibly indicating a common source of (external) bias in the NA models. The error
introduced by external fields is reflected by the bias of the baseline component (LT). For the
period between May and September, the error in modelled ozone due to the boundary
condition is typically small (Solazzo et al., 2012; Im et al., 2015; Giordano et al., 2015;
Hogrefe et al., 2014), while the emissions of ozone precursors and VOCs are problematic,
especially in the EU (Makar et al., 2015; Brunner et al., 2015). We further notice that the
absence of bias in some models may be caused by the presence of compensating bias, i.e.
spatially distributed biases of opposite signs. The spatial distribution of the MSE is discussed
in the next section. In all cases, the $MSE_{best}$ model is, by definition, the model with lowest
MSE and thus the one with the smallest LT bias.
The variance share of LT error is generally small (~1 - 2.5 ppb). This is not entirely
unexpected, as the LT component has a high signal-to-noise ratio with a well-structured
seasonal cycle, peaking in summer. While such a cycle is typically well reproduced by the
models, its phase and/or the amplitude are not always well captured (Solazzo et al., 2012;
Im et al., 2015), leading to the variance error. In detail, the *mMSE* error of the LT component
outweighs the variance error in most cases (in both the EU and NA), and is due to the
unexplained portion of observed variance, thus to the sparseness of the modelled values.
The processes responsible for the *mMSE* error of the LT component (such as deposition,
transport, stratospheric mixing and photochemistry) act at timescales of more than 21 days.
The DU error (on average 3-4 ppb for AQMEII1 and 2-3 ppb for AQMEII2) makes up the
second highest contribution to the total error. The portioning between variance and the
*mMSE* error varies greatly from model to model. However, a comparison of the two AQMEII





phases shows that the *mMSE* is predominant for AQMEII2, while the variance error
(typically due to model under-prediction of the observed variability) is most relevant in
several cases of AQMEII1. Therefore, at the DU scale, the 'quality' of the error of the
AQMEII2 phase is higher than that of its AQMEII1 counterpart. One possible explanation is
the fact that coupled models were used in AQMEII2, while AQMEII1 exclusively used non-
coupled models. As already mentioned (end of section 3), Makar et al. (2015) found that
AQMEII1 models performed better overall with respect to AQMEII2. An analysis of the LT
component showed that the bias in the AQMEII2 models is higher, possibly due to the 2010
emission inventory, while an analysis of the DU error found that the variance error in the
AQMEII2 models is significantly reduced with respect to the AQMEII1 models, and is almost
null. We postulate that the inclusion of feedback effects may have been beneficial, and that
the reduced performance of AQMEII2 models is likely due to external bias. The residual
*mMSE* error of the DU component (~1-2 ppb on average for both continents) is mostly likely
generated by a number of processes, including chemistry, cloudiness, boundary layer
transition and vertical mixing.
The SY error (almost entirely due to *mMSE* in AQMEII2) is comparable across all models
applied to the same continental domain (except for GEM-AQ and WRF/Chem, NA),
indicating that a possible common source of error may be due to missing processes in the
models related to the interaction between chemistry and transport.
Finally, the error of the ID component is less than 1 ppb (on average ~0.2 ppb for AQMEII2)
and is generated by both variance (most commonly model over-prediction) and *mMSE*. The
fast-acting photochemical processes are, therefore, modelled with satisfactory precision.
4.3. SPATIAL DISTRIBUTION OF THE SPECTRAL ERROR COMPONENTS
Maps of MSE by spectral components are reported in Figure 3 to Figure 6. As anticipated by the
error analysis, the LT is the most problematic source of error for both continents, although
the variety in the models' behaviour does not allow for generalisation.
Some of the cases presented in Figure 2, where the bias was null (MM5-CAMx, MM5-DEHM
for AQMEII1 and CosmoArt for AQMEII2, both in EU), show bias compensation, typically due
to model underestimation in the central part of the EU (Germany, eastern France) and
model overestimation in the rest of the continent. The case of the CosmoArt model (Figure 5C)
clearly shows the effect of the spatial averaging in masking the error that is only cancelled
when a continental average is calculated. The model is in fact affected by severe bias and
component errors.
The Po valley in Italy and the southern part of the EU are the most problematic areas,
affected by severe LT errors (Figure 3 and Figure 5). The central and northern parts of the EU are
less problematic, especially for AQMEII2. The other components of the error are
significantly smaller than the LT error, with some exceptions (especially for the DU



component). The length of the segment is in fact normalised to the largest error for each
model, to facilitate the interpretation and the relative weight of each error component.
Concerning NA (Figure 4 and Figure 6), the DU error has more weight and competes with the LT
error in the central and south-eastern parts of the continent. For AQMEII2, the SY error is as
significant as the LT error on the East Coast (Wrf/Chem, Figure 6c). The greatest LT error is
observed in the coastal areas (east and west) and across the north-eastern border between
the US and Canada (due primarily to model underestimation in the east and north, and
model overestimation in the west).
The analysis presented provides a detailed breakdown of the error in terms of error
components, spectral decomposition and spatial distribution, thereby avoiding the pitfalls of
extreme averaging and providing a comprehensive analysis of where the error occurs and
the associated timescales and processes, and whether the error is internally generated or
stems from the model's input data.
5. MSE DECOMPOSITION AND COMPLEXITY
In regression analysis and statistical learning theories, the problem of under- and over-
fitting complex systems is at the root of the MSE decomposition into bias and variance. The
trade-off between bias and variance is strictly dependent on the complexity of the model.
Over-fitting occurs when too many parameters and modules are added to the model: each
new module added to describe a process is a new source of variance due to internal
parameterisation and linearisation. In other words, over-fitting is associated with the
stochasticity inherent to the data/model, and contributes to the increase in variance and
consequent decrease in bias. Under-fitting occurs due to an oversimplification of the
modelled processes, and is an important source of bias as it is associated with the
deterministic property of the modelling activity (Hastie et al., 2009).
The problem of the bias-variance trade-off becomes markedly more complicated when
dealing with complex models with many degrees of freedom, such as AQ modelling systems.
Adding new modules to cope with unexplained physical processes can lead to a reduction in
the bias due to that specific process, but also feeds new variance and possibly new bias into
the model due to the non-linear interaction of the new module with existing ones, since
reducing the bias while preserving the variance is non-trivial. Rao (2005), in the context of
dispersion modelling, provided the theoretical variations of the total model uncertainty by
exploiting the components of the difference between the modelled and observed variance
(Figure 1 of Rao et al., 2005). Rao (2005) used the number of meteorological parameters in
the model as a measure of model complexity, and concluded that the optimal model
complexity could not be defined a priori, but is a trial-and-error combination of the model,
the measurement error and the stochastic uncertainty.
In this study we attempt to derive the curves of the MSE components as a function of model
complexity. Figure 7 shows an example of the approach used to break down model complexity



(which basically relies on the resolved timescale of the model). The complexity of the model
is assumed to increase when the resolved timescale is shortened: the shorter the timescale,
the more complex the model. The timescale of the resolved processes is thus used as a
measure of the complexity, and is obtained by recursively applying the *kz* filter to the ozone
time series. The minimum complexity is assumed to be represented by a model that cannot
resolve any temporal scale below ~1 month (far right of Figure 7), while the maximum
complexity corresponds to the hourly time series, i.e. the standard model's output (far left
of Figure 7).
In Figure 8, we report the spatially averaged curves of bias, variance, and covariance according
to Eq 6 as a function of model complexity. According to the regression analysis theories
outlined above, we would expect the variance to increase according to the complexity
($\frac{d\sigma_m^2}{dcomplexity} > 0$), and the distance between the modelled and observed variance to
decrease $\left(\frac{d(\sigma_m - \sigma_o)^2}{dcomplexity} < 0\right)$, and the opposite for the bias. The curves of variance in Figure 8
indeed turn downwards as predicted by the theory, while the curves of bias have a mixed
behaviour but are, basically, constant $\left(\frac{d(\overline{mod} - \overline{obs})^2}{dcomplexity} \approx 0\right)$. More specifically:
-    The $(\sigma_m - \sigma_o)^2$ term decreases steadily but slowly to a timescale of ~1 day, after
which it drastically drops to significantly lower values. This indicates that *i)* the
complexity of the AQ systems increases exponentially at the DU timescales (not
entirely surprising, given the day/night behavioural properties of ozone); *ii)* the
efforts made to improve the model capabilities on the short-term processes
governing the ozone dynamics improve the model precision; *iii)* there is a possible
lack of parameterisation and modelling of the processes of transport and chemical
transformation over periods longer than 1-2 days.
-    The fact that the bias varies only by small amounts indicates that a fully evolved
model, capable of reproducing processes at the shortest timescales (turbulent
dispersion, fast chemical reactions, even day/night variability, etc.) is no more
accurate than a basic model that only accounts for long-term processes. This might
indicate that *i)* the bias at the shorter timescales is introduced entirely by the larger
timescales, and/or *ii)* the bias is continuously fed into the model by an external
source acting at all scales, as for example the emissions data or boundary conditions.

In Figure S4 to Figure S7 we propose the same analysis as that in Figure 8 but replicated for all
receptors individually (with no spatial average). In most cases (both continents, both
AQMEII phases), the $(\sigma_m - \sigma_o)^2$ term decreases sharply after a timescale of resolved
processes of ~1 day; the bias term confirms the independency to complexity at all
receptors; the covariance is complementary to the variance.
5. CONCLUSIONS



This study presents a novel approach to model evaluation, and aims to combine standard operational statistics with the time allocation of the component error. The methodology we propose tackles the issue of diagnostic evaluation from the angle of the spectral decomposition and error breakdown of model/data signals, introducing a compact operator for the quantification of bias, variance, and the correlation coefficient.

When the analytical decomposition of the error into bias, variance and *mMSE* is applied to the decomposition of the signals into long-term, synoptic, inter-diurnal and diurnal components, information can be gathered that helps reduce the spectrum of possible sources of errors and pinpoint the processes that are most active at a particular scale which need to be improved. The procedure is denoted here as *error apportionment* and provides an improved and more powerful capacity to identify the nature of the error and associate it with a specific part of the spectrum of the model/measurement signal. The AQMEII set of models and measurements have been used in the evaluation procedure.

After analysing the ozone concentrations gathered in the two phases of AQMEII, which cover a number of modelling systems in two different years and geographical areas, we conclude that:

- The bias component of the error is by far the most important source of error, and is mainly associated with long-term processes and/or input fields (likely emissions data or boundary conditions). With regard to the model application, any effort to improve the current capabilities of AQ modelling systems are likely to have little practical impact if this primary issue is not addressed and solved;
- Most relevant to model development, the variance error (the discrepancy between modelled and observed variance) is mainly associated with the DU component. At timescale of ~1-2 days, the complexity of modelling systems increases substantially and many processes are involved; the fact that the variance error of the DU component for the AQMEII2 runs is reduced with respect to the AQMEII1 runs might indicate the benefits of including feedback in the models. Such a conclusion could not be drawn with simpler operational evaluation strategies;
- The limited magnitude of the variability of the SY and LT signals produces little variance errors for these two components, and only becomes comparable to the LT or DU error when the bias is negligible or the total MSE is small;
- The *mMSE* error is predominant in some instances of the analysed models, and is due to the random distribution of modelled values. There are many causes of *mMSE* error, including all 'internal' processes that produce non-systematic errors such as noise, representativeness, the linearisation of non-linear process, and turbulence closure;
- The analysis of the spatial distribution of the error highlights the diversity in the behaviour of each modelling system. The common spatial structures of the LT error (for example in the central and southern EU) may reveal common sources of error




(e.g. emissions data), while the error of the other components (especially DU and SY)
are peculiar to each model and need to be assessed individually.

Analyses of the modelling results for the third phase of AQMEII are currently building on the
methodology outlined in this study, with specific attention being given to the diagnostic of
the error of the LT component in relation to external forcing (emissions and boundary
conditions) and of the DU component with respect to the variance error.



APPENDIX
As in Hogrefe et al. (2000) and Galmarini et al. (2013), the time windows ($m$) and the
smoothing parameter ($k$) have been selected as follows:

$$ID(t) = \mathbf{x}(t) - kz_{3,3}(\mathbf{x}(t))$$
$$DU(t) = kz_{3,3}(\mathbf{x}(t)) - kz_{13,5}(\mathbf{x}(t))$$
$$SY(t) = kz_{13,5}(\mathbf{x}(t)) - kz_{103,5}(\mathbf{x}(t))$$
$$LT(t) = kz_{103,5}(\mathbf{x}(t))$$
$$\mathbf{x}(t) = ID(t) + DU(t) + SY(t) + LT(t)$$

EQ. S.1

where $\mathbf{x}(t)$ is the time series vector.
A clear-cut separation of the components of EQ. S.1 cannot be achieved, as the separation is
a non-linear function of the parameters $m$ and $k$ (Rao et al., 1997). It follows that the
components of EQ. S.1 are not completely orthogonal and that some level of overlapping
energy exists (Kang et al., 2013). Galmarini et al. (2013) found that the explained variance by
the spectral components account for 75 to 80% of the total variance, the remaining portion
being explained by the interactions between the components.

Assuming a spectral decomposition which is valid for the modelling and the observational
time series, the MSE formulation outlined in Galmarini et al. (2013) holds:

$$MSE(O3) = MSE(LT + SY + DU + ID)$$
$$= \sum MSE(spec\ comp) + \sum MSE(cross\ comp)$$

EQ. S.2


Where *spec comp* are the diagonal terms, and *LT, SY, DU, ID* and *cross comp* are the off-
diagonal terms deriving from the squared nature of the MSE: $LT_oSY_m$, $SY_oLT_m$, $SY_oDU_m$,





$DU_oSY_m$, $DU_oID_m$, $ID_oDU_m$, $LT_mSY_m$,   $LT_oSY_o$, $DU_mSY_m$, $DU_mID_m$, $DU_oSY_o$, $DU_oID_o$ (*o* and *m*
represent observed and modelled fields, respectively). For simplicity, the cross-components
are assumed to be symmetric, so the *o* and *m* subscripts are dropped. This simplification has
little impact on the MSE breakdown since, as shown by Galmarini et al. (2013), the diagonal
terms alone account for over 80% of the total variance.
To isolate the contribution to MSE of a single spectral component, we proceed as follows.
We subtract a component (e.g. LT) from the whole time series:

$$MSE(O_3\text{-}LT(O_3)) =$$
$$MSE(SY)+MSE(DU)+MSE(ID)+2MSE(IDDU)+2MSE(IDSY)+2MSE(DUSY)$$

EQ. S.3


By removing EQ. S.3 from EQ. S.2, the contribution of LT and its cross-component is isolated:

$$EQ.\ S.2\text{-}\ EQ.\ S.3 = MSE(LT) + MSE(LTID) + MSE(LTSY) + MSE(LTDU)$$

EQ. S.4


We can further elaborate on EQ. S.4 to isolate the contribution of each cross-component.
For instance, the case of SYLT:

$$MSE(SY\text{-}ID\text{-}DU)\text{–}MSE(SY)\text{–}MSE(LT) = [MSE(SY)+MSE(LT)+ 2MSE(SYLT)] - MSE(SY) -$$
$$MSE(LT) = 2MSE(SYLT)$$

EQ. S.5


The procedure in EQ. S.5 has been applied to derive the contribution of all cross-
components.

ACKNOWLEDGEMENTS
We would like to thank the community of modellers and data providers of the first and
second phases of AQMEII.








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



FIGURES
**Figure 1.** Share (in %) of the total MSE in the main spectral components and the cross components (see Appendix for
detail) for *a)* AQMEII1 and *b)* AQMEII2. Top panel: EU; lower panel: NA.
**Figure 2.** MSE (ppb$^2$) breakdown in bias, variance and mMSE of the spectral components ID, DU, SY, LT, based on Eq 9. The
sign within the share of bias and variance indicates model overestimation (+) or underestimation (-) of mean concentration
(bias) and variance. *a)* AQMEII1 and *b)* AQMEII2. Top panel: EU; lower panel: NA.
**Figure 3** Spatial distribution of the MSE in the spectral components for the EU models of AQMEII1. The segments are
centred at the rural receptors' position (clockwise from north: MSE of ID, DU, SY, and LT). Their length is proportional to
the MSE magnitude, coded according to the colour scale. For each model, the colour scale extends from zero up to the 75$^{th}$
percentile, and the last value of the scale is the maximum MSE. The colour of the MSE values above the 75$^{th}$ percentile
represents the maximum value. The thick dashed LT segment indicates model underestimation (low model bias).
**Figure 4** As in **Figure 3,** but for the NA models of AQMEII1.
**Figure 5**. As in **Figure 3,** but for the EU models of AQMEII2.
**Figure 6** As in **Figure 3,** but for the NA models of AQMEII2.
**Figure 7** Example of the model complexity as time-resolved scale of the transport and dispersion processes: the minimum
complexity (far right) is a poor time-resolving time series obtained as kz(250,5). The complexity increases towards the left,
with the scale of resolved processes becoming finer up to the maximum complexity (far left), which represents the full time
series.
**Figure 8** Evolution of error components (Red: bias; Blue: variance; Black: covariance) as a function of model complexity.
Complexity increases from left (min.) to right (max.) and is calculated as the temporal scale of the resolved process using
the kz filter on the modelled signal: kz(i,5), i=2,…,250.













TABLES
**Table 1.** Features of the modelled domains

|  | Europe | | North America | |
|---|---|---|---|---|
|  | **phase 1** | **phase 2** | **phase 1** | **phase 2** |
| **Simulated year** | 2006 | 2010 | 2006 | 2010 |
| **Extension** | (-10,39)W; (30,65)N | | (-125,-55)W; (26,51)N | |
| **Number of receptors** (min validity=75%; max altitude = 1000 m) | 1339 | 1360 | 672 | 652 |
























**Table 2**. Modelling systems participating in the first (Table a) and second (Table b)  phases of AQMEII for Europe and North
America
*a)*

| Model | | | Grid(km) | Emissions | Chemical BC |
|---|---|---|---|---|---|
| Code | Met | AQ | | | |
| **EUROPE** | | | | | |
| DK1 | MM5 | DEHM | 50 | Global emission databases, EMEP | Satellite measurements |
| FR3 | MM5 | Polyphemus | 24 | Standard§ | Standard |
| HR1 | PARLAM-PS | EMEP | 50 | EMEP model | From ECMWF and forecasts |
| UK2 | WRF | CMAQ | 18 | Standard§ | Standard |
| US4 | WRF | WRF/Chem | 22.5 | Standard§ | Standard |
| FI1 | ECMWF | SILAM | 24 | Standard anthropogenic; In-house biogenic | Standard |
| FR4 | MM5 | Chimere | 25 | MEGAN, Standard | Standard |
| PL1 | GEM | GEM-AQ | 25 | Standard over AQMEII region; Global EDGAR/GEIA over the rest of the global domain | Global variable grid setup (no boundary conditions) |
| NL1 | ECMWF | Lotos-EUROS | 25 | Standard§ | Standard |
| DE1 | COSMO | Muscat | 24 | Standard§ | Standard |
| US3 | MM5 | CAMx | 15 | MEGAN, Standard | Standard |
| DE3 | COSMO-CLM | CMAQ | 24 | Standard§ | Standard |
| **NORTH AMERICA** | | | | | |
| CA1 | GEM | AURAMS | 45 | Standard* | Climatology |
| PL1 | GEM | GEM-AQ | 25 | Standard over AQMEII region; Global EDGAR/GEIA over the rest of the global domain | Global variable grid setup (no boundary conditions) |
| PT1 | MM5 | CAMx | 24 | Standard | LMDZ-INCA |
| US1 | WRF | CAMQ | 12 | Standard | Standard |
| US3 | WRF | CAMx | 12 | Standard | Standard |
| FR4b | WRF | CHIMERE | | | |
| DK1 | MM5 | DEHM | 50 | Global emission databases, EMEP | Satellite measurements |
| DE3 | COSMO-CLM | CMAQ | 24 | Standard§ | Standard |
| ES3 | WRF | WRF/Chem | 23 | Standard | Standard |

§ Standard anthropogenic emissions and biogenic emissions derived from meteorology (temperature and solar radiation) and land use
distribution implemented in the meteorological driver.





*Standard anthropogenic inventory but independent emission processing, exclusion of wildfires, and different versions of BEIS(v3.09)
used.
Refer to Solazzo et al. (2012a-b) and references therein for details.
*b)*

| Model | | | Grid | Emissions | Chemical BC |
|---|---|---|---|---|---|
| Code | Met | AQ | | | |
| **EUROPE** | | | | | |
| AT1 | WRF | WRF/Chem | 23 km | Standard | Standard |
| CH1 | COSMO | Cosmo-ART | 0.22° | Standard | Standard |
| ES2a | NMMB | BSCCTM | 0.20° | Standard | Standard |
| ES3 | WRF | WRF/Chem | 23 km | Standard | Standard |
| NL2 | RACMO | LOTOS-EUROS | 0.5° x 0.25° | Standard | Standard |
| UK5 | WRF | CMAQ | 18 km | Standard | Standard |
| UK4 | MetUM | UKCA RAQ | 0.22° | Standard | Standard |
| DE3 | COSMO | Muscat | 0.25° | Standard | Standard |
| **NORTH AMERICA** | | | | | |
| ES1 | WRF | WRF/CHem | 36 km | Standard | Standard |
| US6 | WRF | CMAQ | 12km | Standard | Standard |
| CA2f | GEM | MACH | 15 km | Standard | Standard |

Standard Boundary conditions: 3-D daily chemical boundary conditions were provided by the ECMWF IFS-MOZART model run in the
context of the MACC-II project (Monitoring Atmospheric Composition and Climate - Interim Implementation) at 3-hourly and 1.125 spatial
resolution. Refer to Im et al. (2015a-b) for details.
Standard Emissions: based on the TNO-MACC-II (Netherlands Organization for Applied Scientific Research, Monitoring Atmospheric
Composition and Climate - Interim Implementation) framework for Europe and by the US EPA (Environmental Protection Agency) and
Environment Canada for North America. The 2008 National Emissions Inventory (http://www.epa.gov/ttn/chief/net/2008inventory.html)
and the 2008 Emissions Modeling Platform (http://www.epa.gov/ttn/chief/ emch/index.html#2008) with year-specific updates for 2006
and 2010 were used for the US portion of the modelling domain. Canadian emissions were derived from the Canadian National Pollutant
Release Inventory (http://www.ec.gc.ca/inrp-npri/) and Air Pollutant Emissions Inventory (http://www.ec.gc.ca/inrp-npri/ donnees-
data/ap/index.cfm?lang¼En) values for the year 2006. Refer to Im et al. (2015a-b) for details.





ERROR APPORTIONMENT FOR ATMOSPHERIC CHEMISTRY-TRANSPORT MODELS. A NEW APPROACH TO
MODEL EVALUATION, BY E. SOLAZZO, S. GALMARINI

FIGURES

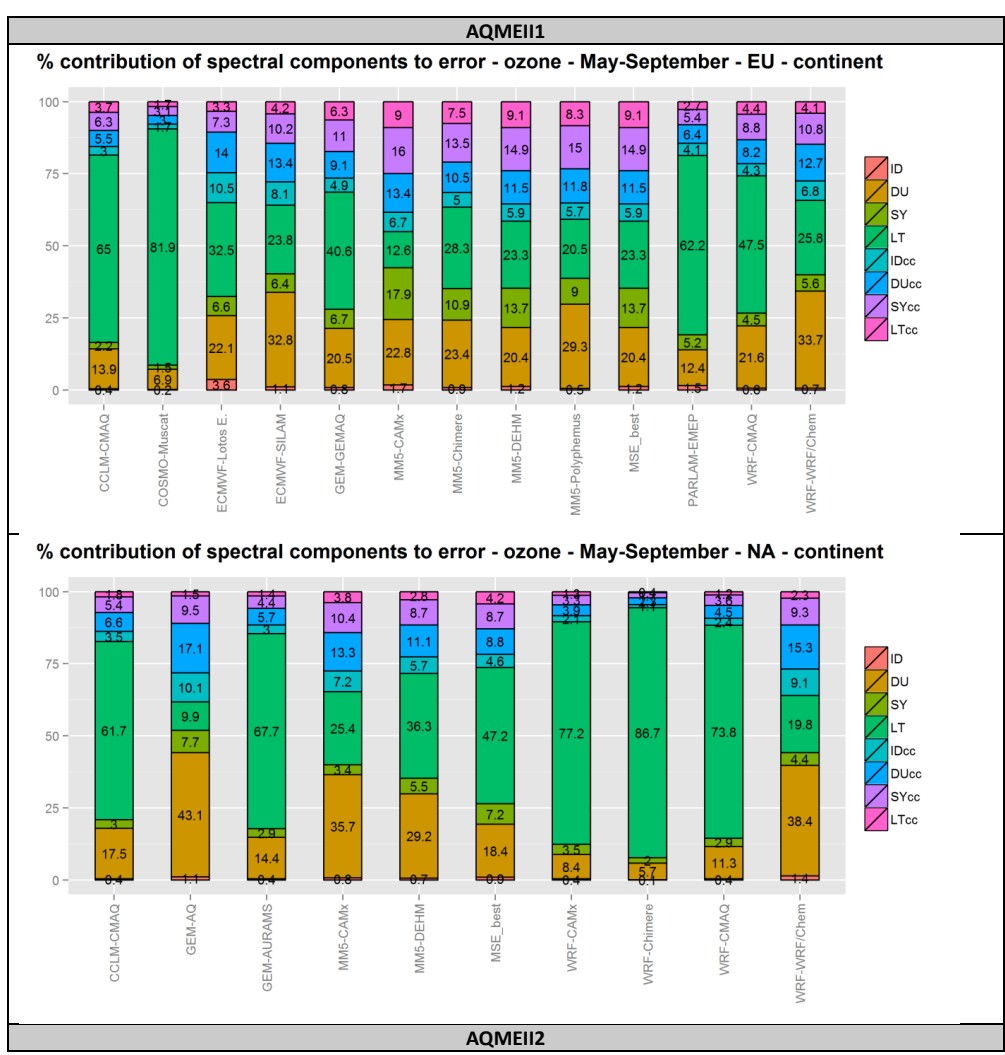





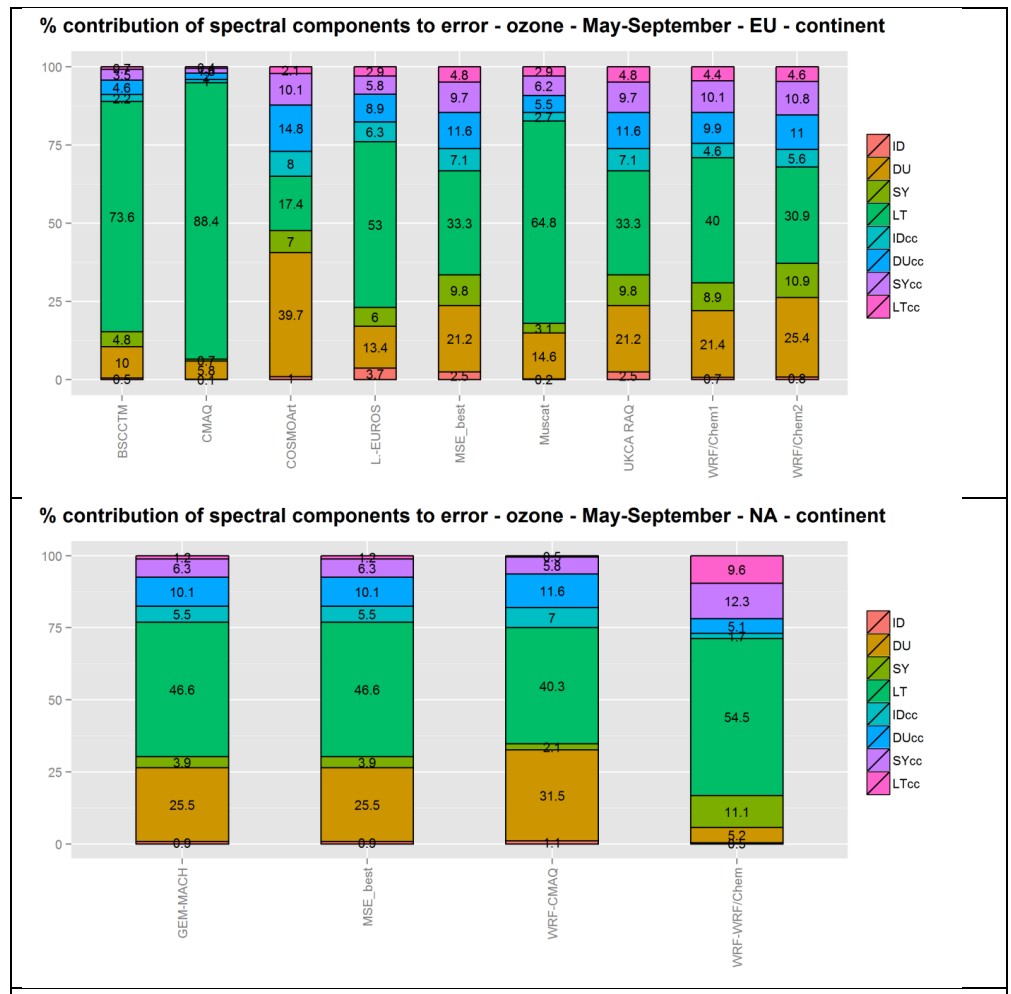

**Figure 1.** Share (in %) of the total MSE in the main spectral components and the cross components (see Appendix for detail) for a) AQMEII1 and b) AQMEII2. Top panel: EU; lower panel: NA.





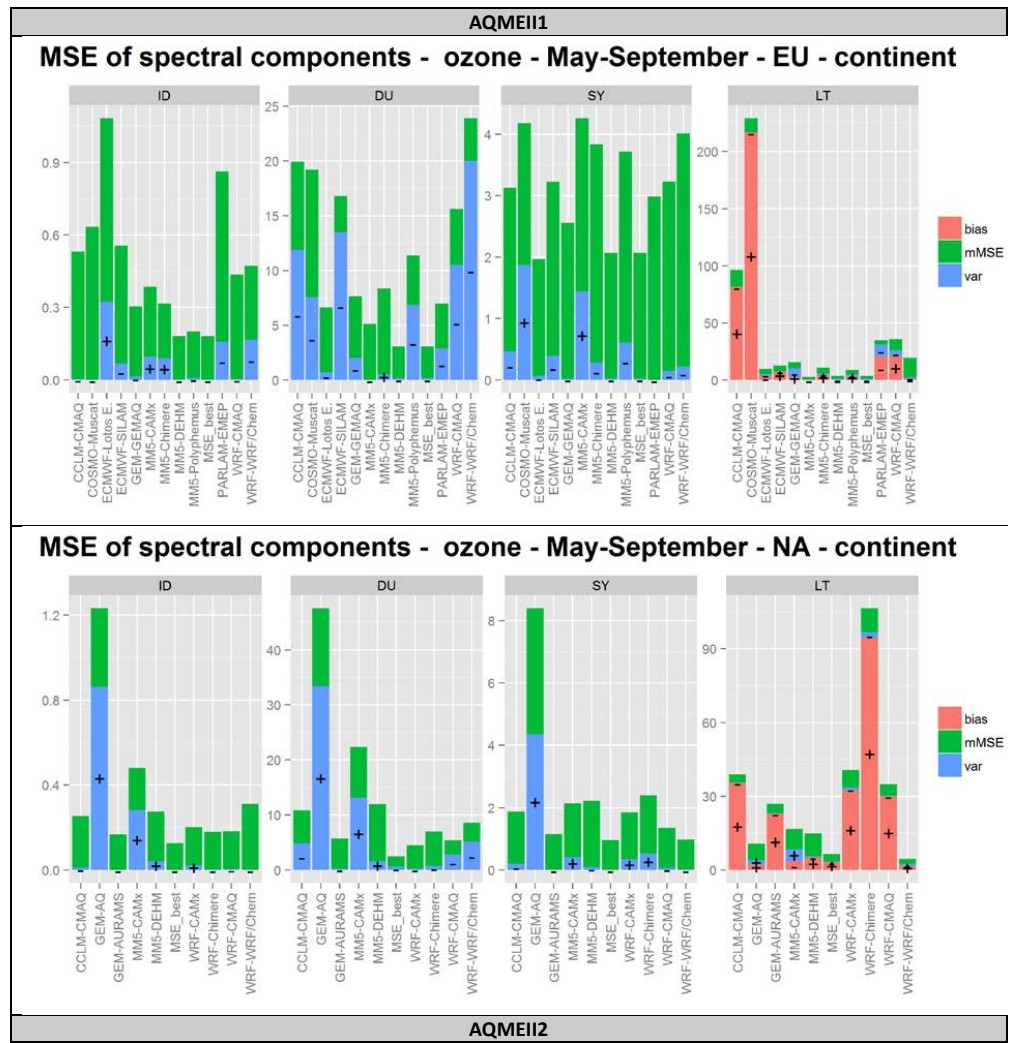





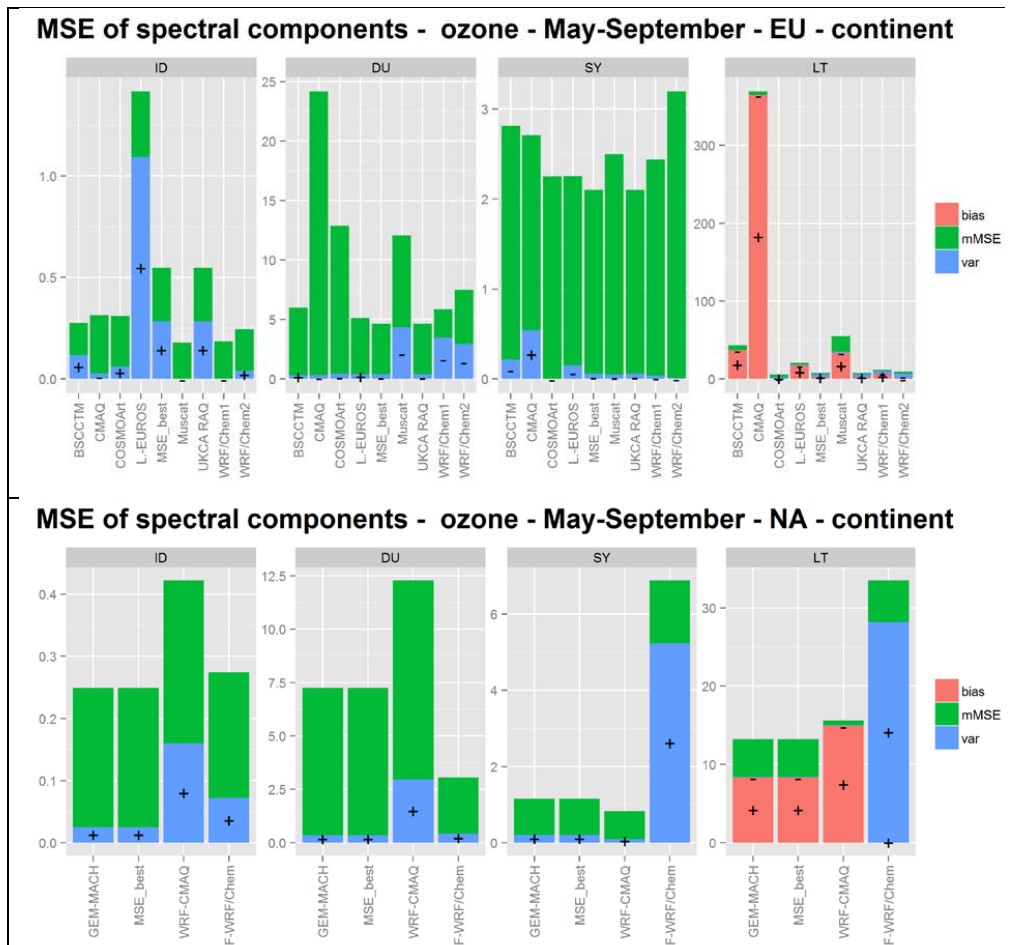

**Figure 2.** MSE (ppb2) breakdown in bias, variance and mMSE of the spectral components ID, DU, SY, LT, based on Eq 9. The sign within the share of bias and variance indicates model overestimation (+) or underestimation (-) of mean concentration (bias) and variance. a) AQMEII1 and b) AQMEII2. Top panel: EU; lower panel: NA.





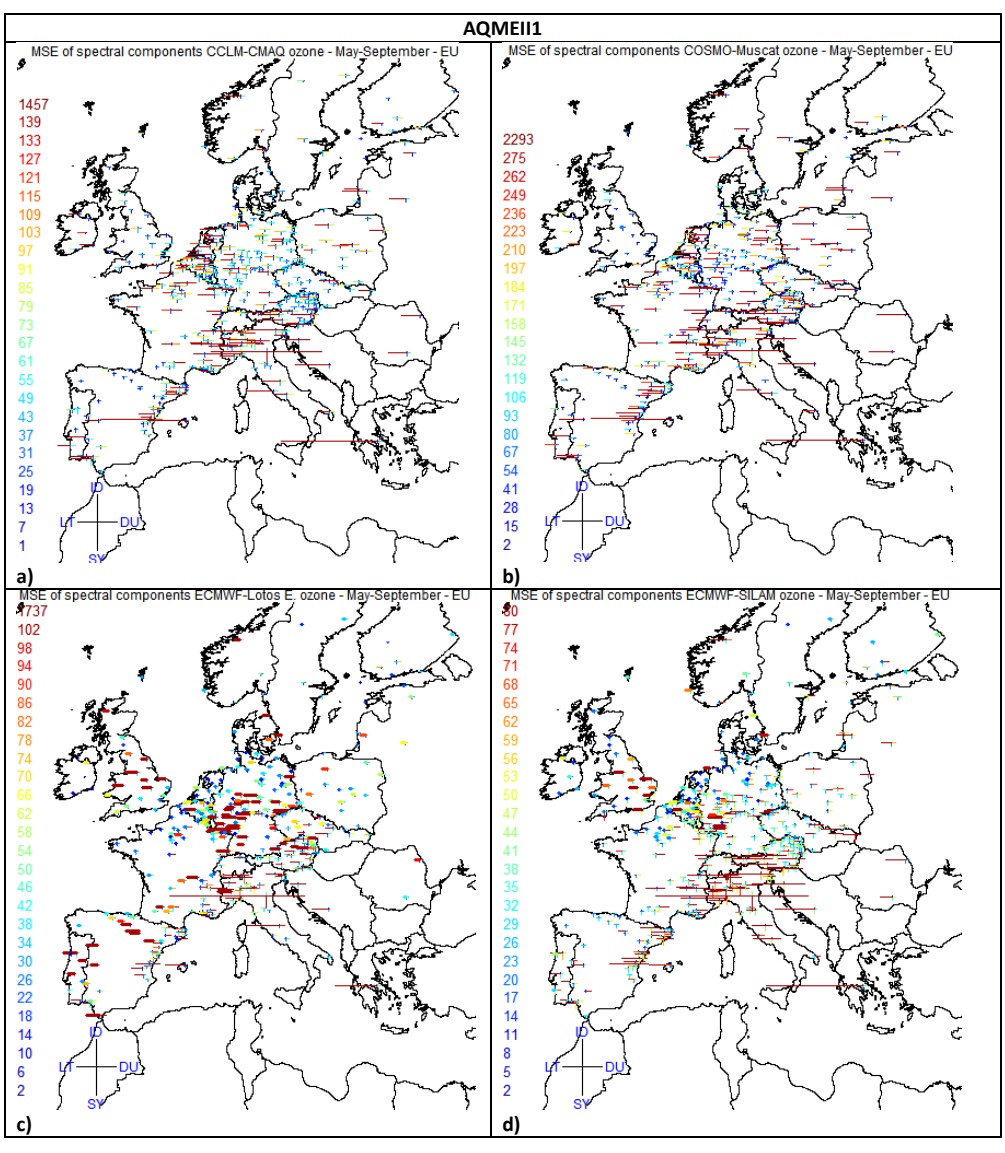



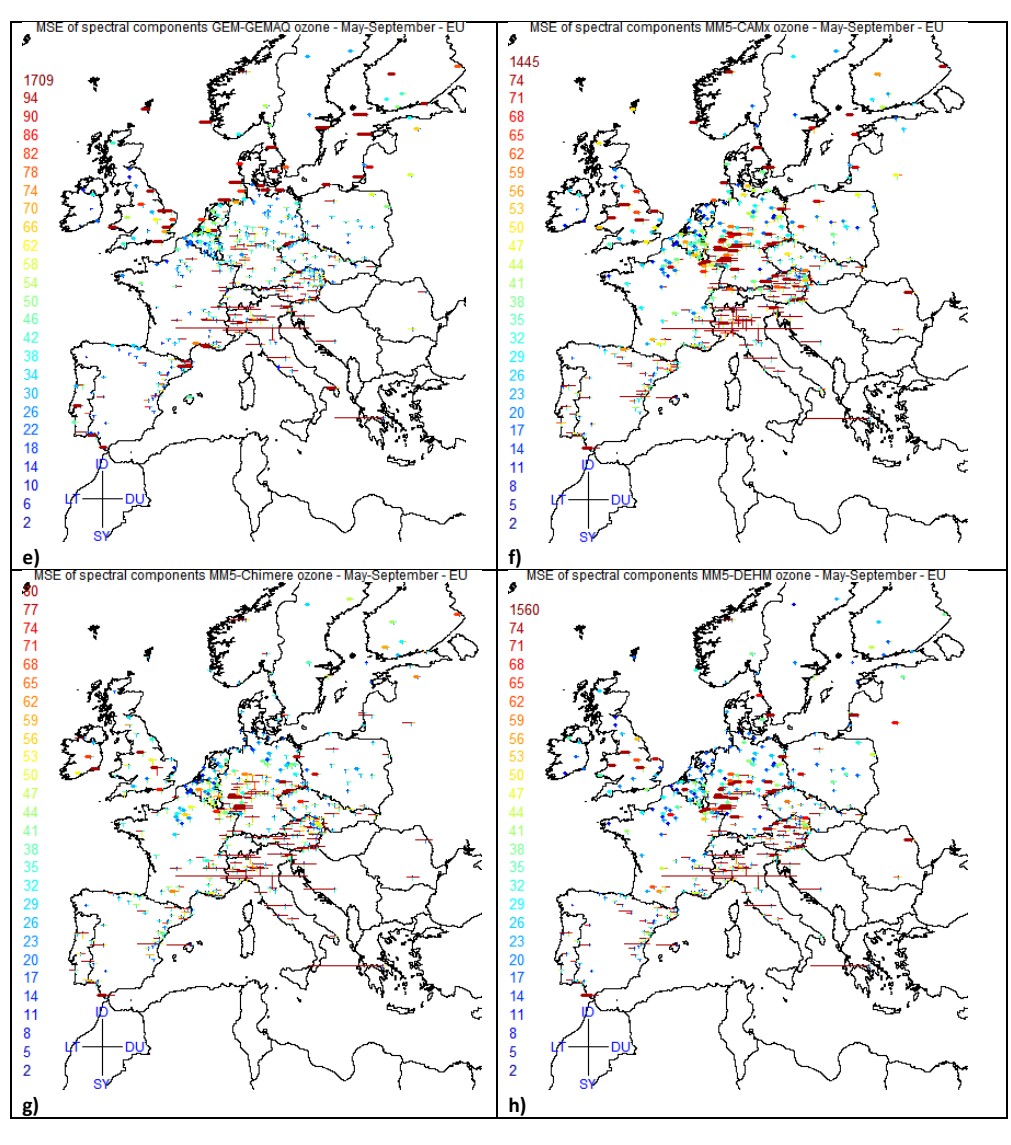





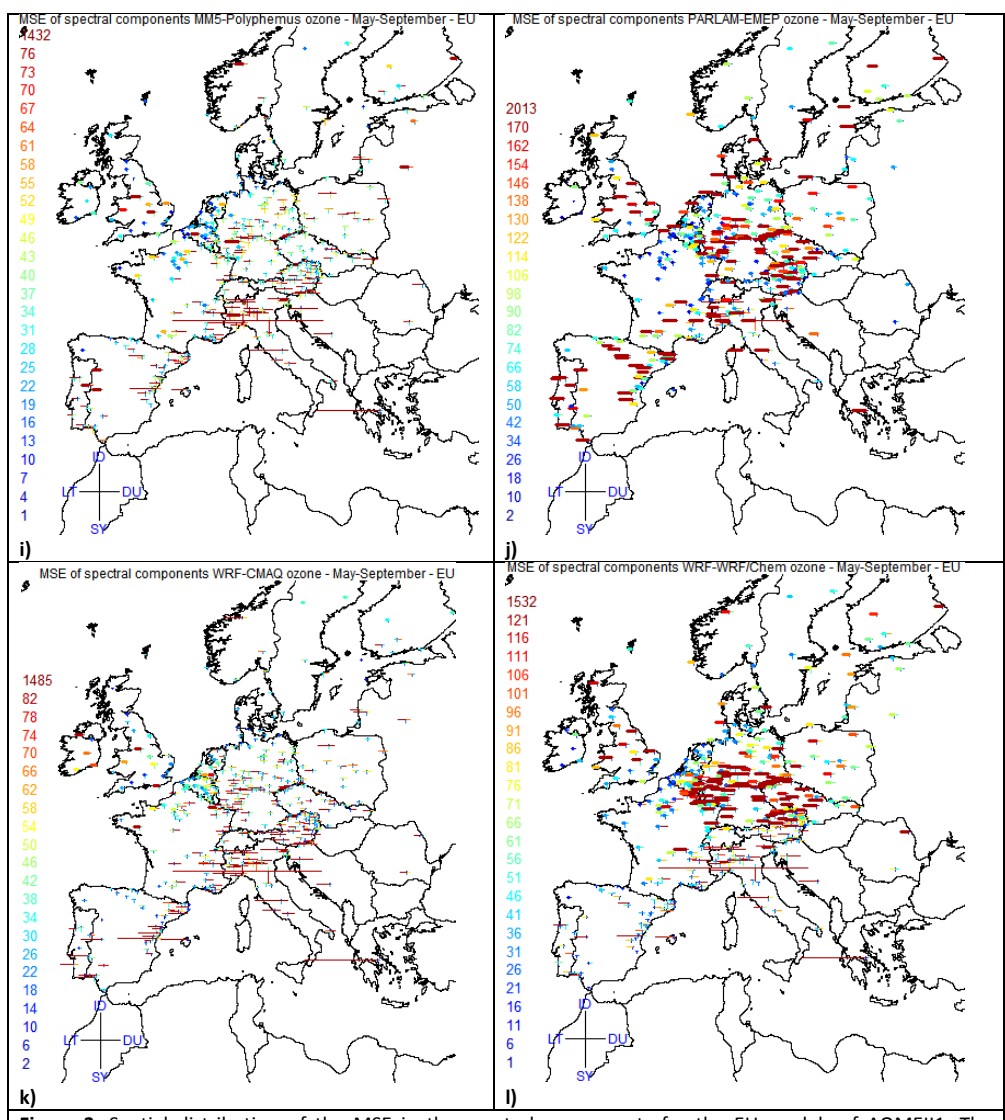

**Figure 3**. Spatial distribution of the MSE in the spectral components for the EU models of AQMEII1. The segments are centred at the rural receptors' position (clockwise from north: MSE of ID, DU, SY, and LT). Their length is proportional to the MSE magnitude, coded according to the colour scale. For each model, the colour scale extends from zero up to the 75th percentile, and the last value of the scale is the maximum MSE. The colour of the MSE values above the 75th percentile represents the maximum value. The tick-dashed LT segment indicates model underestimation (low model bias).













**Figure 4**. As in **Figure 3** but for the NA models of AQMEII1







**Figure 5**. As in **Figure 3** but for the EU models of AQMEII2



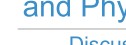

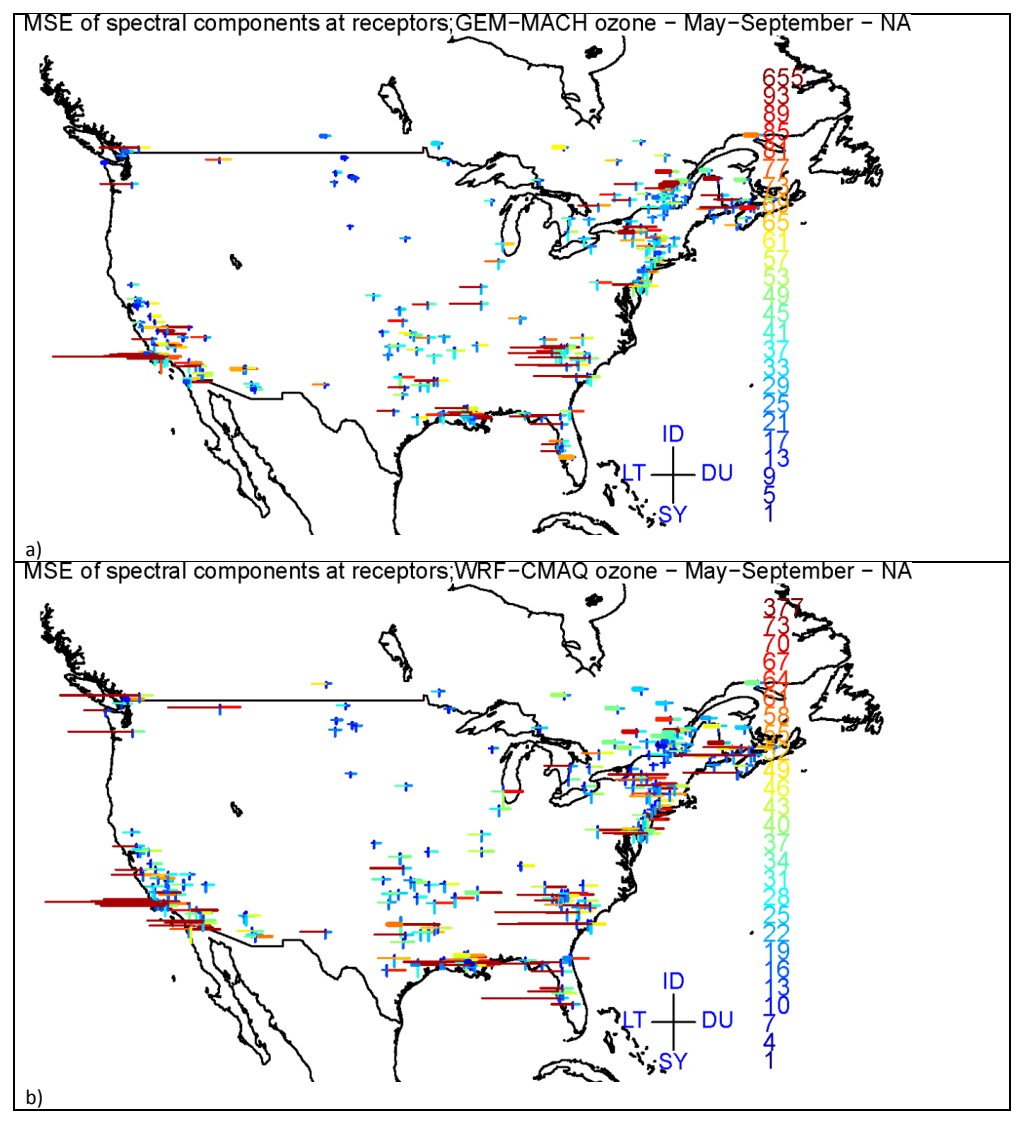





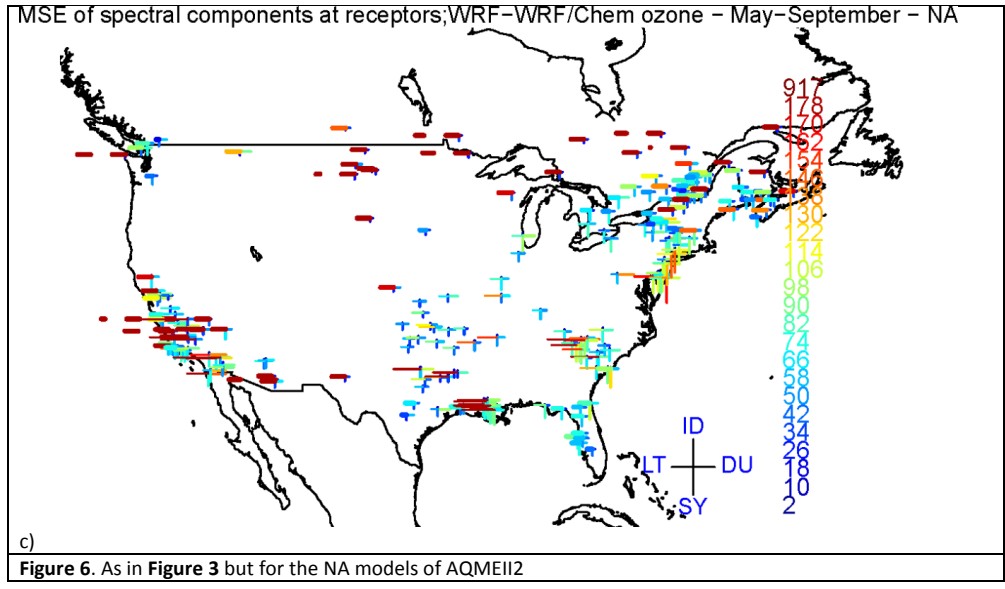

**Figure 6**. As in **Figure 3** but for the NA models of AQMEII2





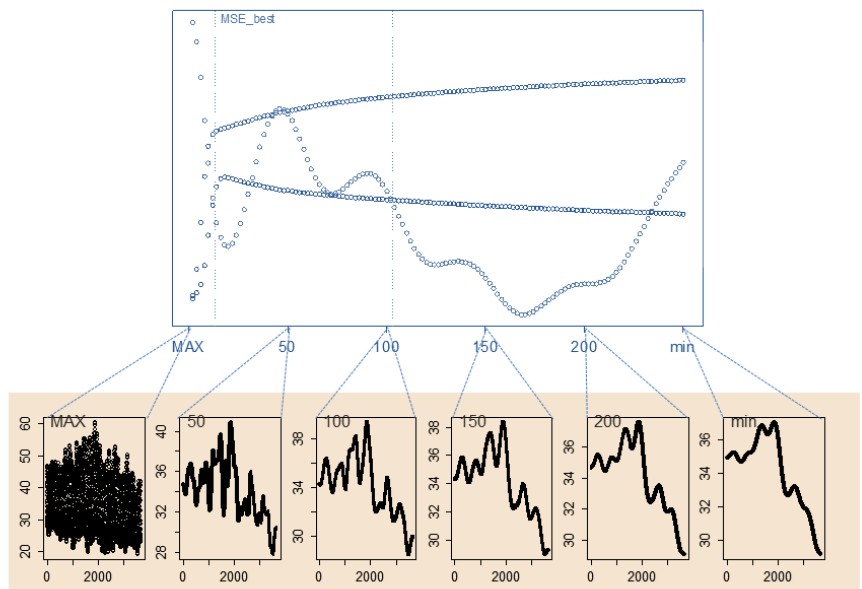

**FIGURE 7** Example of the model complexity as time-resolved scale of the transport and dispersion processes: the minimum complexity (far right) is a poor time-resolving time series obtained as kz(250,5). The complexity increases towards the left, with the scale of resolved processes becoming finer up to the maximum complexity (far left), which represents the full time series.





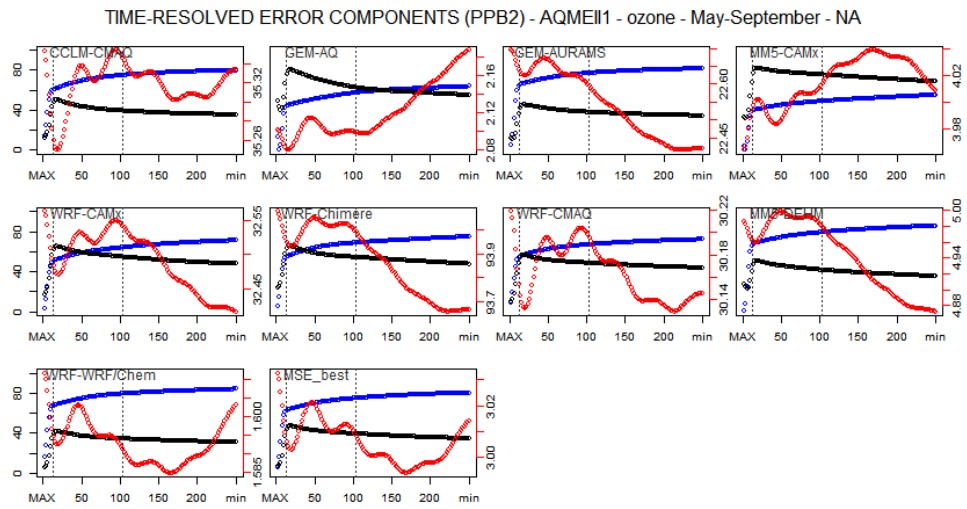





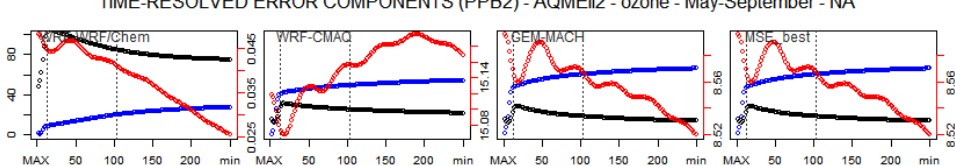

**FIGURE 8** Evolution of error components (red: bias; Blue: variance; Black: covariance) as a function of model complexity. Complexity increases from left (min.) to right (max.) and is calculated as the temporal scale of the resolved process using the kz filter on the modelled signal: kz(i,5), i=2,…,250.