# Peer review of "ERROR APPORTIONMENT FOR ATMOSPHERIC CHEMISTRY-TRANSPORT"

_Atmospheric Chemistry and Physics, 2016_

## Referee Comment (RC1) · Anonymous Referee #2 · 22 Mar 2016

General comments

The work presented herein, presents a new approach to model evaluation that attempts to shade light into the processes that influence model errors, rather than traditionally compare modeled ozone concentrations to in-situ measured values. The methodology is scientifically solid and sound and will help the AQ community move toward new ways of error diagnostics and thus, model improvement. The title of the manuscript reflects the contents of the paper and is considered sufficient. The main comments from the review process are related to obscure parts in the discussion of the figures and results. The specific comments that follow are meant to strengthen the communication of the results to readers that may not be as familiar with the history of the AQMEII initiative or

the details of the spectral decomposition methodology. I am in favor of publishing this paper with Atmospheric Chemistry and Physics, after addressing the minor specific comments that follow.

Specific comments/suggestions

1. Section 2.1: In the beginning of the error decomposition section, please add references to the original published work (i.e. Wilmott, Murphy and others). The part that was uniquely developed for this work should also be clearly identified in this section.

2. Page 5: in the minimization of MSE, the authors want to achieve independency of MSE from the model's statistical metrics, since the observed ones are not controllable. Can you please add a brief explanation in the text as to why you chose to differentiate over the mean model value and model standard deviation?

3. Page 6, spectral decomposition: In Rao et al. (1997) and Hogrefe et al. (2000) the ozone time series are log-transformed before the analysis to stabilize the variances. Did the authors use the log-transform in their KZ application? If not, please explain the rationale behind using the original ozone data.

4. Page 6, lines 188-190: what is the meaning of the bias in the discussion of the decomposition components? Equation 10 is applied to modeled and observed values separately.

5. Table 2: please denote in the title which table corresponds to which AQMEII phase.

6. Page 8, section 4.1, line 249: the phrase "spatially averaged over the two continental areas" must be rephrased to "spatially averaged over each continental area". I am assuming that the MSE is calculated for each spectral component and each station and then averaged over each continent (there is one value of MSE for each component and each station for the period of May-Sep). Please clarify in the text accordingly.

7. In Figure 1, the cross components are denoted by subscript cc in the name of the variable. I suggest using the same name convention in the appendix, where the

description of the cross components is included. This will avoid confusion to readers that are not familiar with the prior literature.

8. Page 9, lines 281-283: Is this statement based on results from the current or previous published work? Please add a reference to this statement accordingly.

9. Page 9: in the 1st paragraph of section 4.2 the bias is described as influenced by both internal and external model errors (which is true). In the 2nd paragraph (line 292), the authors suggest that the bias of LT shows the externally induced errors. Can you clarify this inconsistency?

10. The units in Figure 2 are ppb square (ppb2) or ppb? If the former is true, then the MSE breakdown must be bias2, variance and mMSE from equation 9? Please revise the label accordingly.

11. Section 4.3, figures 3-6: even though I embrace the idea of including a lot of information in one plot, it has been very challenging to read and understand the figures. I don't understand where the under- or over-estimation is indicated. I suggest the inclusion of one example (maybe in the figure caption or in the text) that will describe the results from one specific station (highlighted with a square of circle). That way, it will be easier for the reader to connect the color coded scale with the different components. The plots provide valuable information which must be communicated in the most efficient manner.

12. Page 12, figure 7: the components of figure 7 must be explained in more detail. What are the units? x and y axes? What is shown in the upper plot? The paragraph describing figure 7 and the method behind it needs further improvement to communicate a clear message.

———————————————————————

---

## Referee Comment (RC2) · Anonymous Referee #1 · 25 Mar 2016

Comments

This is an interesting and well written paper that makes a meaningful contribution to model evaluation. A few comments and editorial suggestions are provided below.

Wavelet filters can provide better separation of components (i.e., reduced covariances among components).

The spatial support of the model (model grid average) is greater than that of the observations (point scale), and should therefore have a smaller variance, as should all the temporal components. The term $\sigma_m$ will typically be less than $r\sigma_o$ for this reason.

The model/observation agreement in the DU component is driven largely by diurnal

forcing (similarly, the LT component has a significant amount of annual energy). Model performance metrics for the DU component is misleadingly optimistic because it mostly reflects the 24 hour and annual forcings embedded in both the observations and model values. For periodic processes, metrics derived from the amplitude and phase can be more informative.

The variance of the ID term is very small. Therefore, although the paper shows both the fraction of variance due to each component and the error terms, it should be pointed out that the small errors in the ID component are quite large relative to the total amount of ID variability.

Model/observation correlation as a stand-alone metric can be informative as it shows whether the model can reproduce patterns seen in the observations. For example, the ID component, as noted, has small errors, but for individual monitoring sites (not spatially averaged), correlation between modeled and observed ID is often quite low and insignificant (there often appears to be no relationship between the two). On the other hand, correlation tends to improve as time and space scales increase, often leaving the LT component with the best agreement in terms of correlation.

  Editorial comments

There is some confusion in the text when discussing bias. Figure 2 actually shows squared bias, though the discussion seems to be referring to both bias and squared bias.

Line 263: should read "has little impact" or "has negligible impact"

Line 283: The statement ending on this line could use a reference.

Line 305: What is meant by "sparseness of the modeled values"?

Line 452: should have a period at the end

Line 457: should have a period at the end

Figure 1. Panels do not have 'a)' and 'b)' labels. Also, if it's not too much trouble, invert the legends so that the colors appear in the same order as they do in the bars.

Figures 3 and 4 are very difficult to look at. When error terms are small it is hard to tell where the intersection is. Zooming in would help with better image resolution.

Figure 8: Caption should read 'from right to left'.

[Figure]

---

## Author Comment (AC1) · 19 Apr 2016

**General comments.**

The work presented herein, presents a new approach to model evaluation that attempts to shade light into the processes that influence model errors, rather than traditionally compare modeled ozone concentrations to insitu measured values. The methodology is scientifically solid and sound and will help the AQ community move toward new ways of error diagnostics and thus, model improvement. The title of the manuscript reflects the contents of the paper and is considered sufficient. The main comments from the review process are related to obscure parts in the discussion of the figures and results. The specific comments that follow are meant to strengthen the communication of the results to readers that may not be as familiar with the history of the AQMEII initiative or the details of the spectral decomposition methodology. I am in favor of publishing this paper with Atmospheric Chemistry and Physics, after addressing the minor specific comments that follow.

**Specific comments/suggestions**

1. Section 2.1: In the beginning of the error decomposition section, please add references to the original published work (i.e. Wilmott, Murphy and others). The part that was uniquely developed for this work should also be clearly identified in this section.

**Response.** The content of section 2.1 actually reflects the origin of the methodology we propose, which is derived from many fields and never applied to air quality (or geophysical time series, to our knowledge) before. This is the first work that put together the Theil decomposition and the minimisation of the error for spectrally decomposed time series. We have moved the reference to Murphy (1988) at the beginning of the section, but rather keep the rest unchanged.

2. Page 5: in the minimization of MSE, the authors want to achieve independency of MSE from the model's statistical metrics, since the observed ones are not controllable. Can you please add a brief explanation in the text as to why you chose to differentiate over the mean model value and model standard deviation?

**Response. Explanation added to the text**

3. Page 6, spectral decomposition: In Rao et al. (1997) and Hogrefe et al. (2000) the ozone time series are log-transformed before the analysis to stabilize the variances. Did the authors use the log-transform in their KZ application? If not, please explain the rationale behind using the original ozone data.

**Response.** We used the original time series of ozone data. Prior to the analysis, tests have shown that the results of the MSE breakdown were independent from the log transform of the initial data. We have used an approach consistent with Galmarini et al. (2013), where the raw data were also used.

4. Page 6, lines 188-190: what is the meaning of the bias in the discussion of the decomposition components? Equation 10 is applied to modeled and observed values separately.

**Response.** The bias should intended as presented in section 2.1, as from from Johnson et al (2008): 'the closeness of agreement between the average value obtained from a large series of measurements and the true value', where the keyword is 'average'. The bias is the off-set of the averaged model results from the averaged measured values. In this sense, the band-pass components ID, DU, SY have zero mean by definition, and are therefore unbiased.

5. Table 2: please denote in the title which table corresponds to which AQMEII phase.

**Response. Done**

6. Page 8, section 4.1, line 249: the phrase "spatially averaged over the two continental areas" must be rephrased to "spatially averaged over each continental area". I am assuming that the MSE is calculated for each spectral component and each station and then averaged over each continent (there is one value of MSE for each component and each station for the period of May-Sep). Please clarify in the text accordingly.

**Response. Done, it has been clarified in the text**

7. In Figure 1, the cross components are denoted by subscript cc in the name of the variable. I suggest using the same name convention in the appendix, where the description of the cross components is included. This will avoid confusion to readers that are not familiar with the prior literature.

**Response. Done**

8. Page 9, lines 281-283: Is this statement based on results from the current or previous published work? Please add a reference to this statement accordingly.

**Response.** The statement is not derived from previous studies but based on the experience of the current work.**

9. Page 9: in the 1st paragraph of section 4.2 the bias is described as influenced by both internal and external model errors (which is true). In the 2nd paragraph (line 292), the authors suggest that the bias of LT shows the externally induced errors. Can you clarify this inconsistency?

**Response.** The inconsistency is driven by the word 'error' rather than 'bias'. It has been corrected now. All biases (internal and external) are driven by the LT component, thus it is correct to say that the bias of the external drivers is incorporated in the bias of the LT component.

10. The units in Figure 2 are ppb square (ppb2) or ppb? If the former is true, then the MSE breakdown must be bias2, variance and mMSE from equation 9? Please revise the label accordingly.

**Response. We have added labels to figure 2 and modified the caption accordingly**

11. Section 4.3, figures 3-6: even though I embrace the idea of including a lot of information in one plot, it has been very challenging to read and understand the figures. I don't understand where the under- or overestimation is indicated. I suggest the inclusion of one example (maybe in the figure caption or in the text) that will describe the results from one specific station (highlighted with a square of circle). That way, it will be easier for the reader to connect the color coded scale with the different components. The plots provide valuable information which must be communicated in the most efficient manner.

**Response.** Thanks for the valuable suggestion. We have improved the readability of the figure and added, as an example, a scheme on how the figure has to be interpreted (see last panel of figure 3).**

12. Page 12, figure 7: the components of figure 7 must be explained in more detail. What are the units? x and y axes? What is shown in the upper plot? The paragraph describing figure 7 and the method behind it needs further improvement to communicate a clear message.

**Response.** The figures 7 and 8 have been revised and improved. We have also slightly revised the contents of Section 5, which has already a detailed and independent introduction, with examples and review of the results. We acknowledge the topic might be not straightforward to understand and therefore have made extra effort in trying to simplify it.

**Anonymous Referee #1**

This is an interesting and well written paper that makes a meaningful contribution to model evaluation. A few comments and editorial suggestions are provided below.

1. Wavelet filters can provide better separation of components (i.e., reduced covariances among components).

**Response.** Eskridge et al. (1997) compared the kz filter against several other methods, including wavelet filters, showing that kz has the same level of accuracy and (often) higher level of confidence. The kz filter has also the advantage of 1. being insensitive to missing values, 2. being supported by extensive literature when applied to ozone, 3. depending on two parameters only, which are quite robust for ozone. It is true, however, that the main shortcoming of method we have developed is the overlapping between the cross components and especially the fact that the error of cross components can be quantified but cannot be apportioned according to the methodology outlined in the current work. Nonetheless, we have preferred to rely on this methodology and possibly exploring wavelet in the future.

2. The spatial support of the model (model grid average) is greater than that of the observations (point scale), and should therefore have a smaller variance, as should all the temporal components. The term om will typically be less than roo for this reason.

**Response. We have included some comments in the text (see line 306 onwards)**

3. The model/observation agreement in the DU component is driven largely by diurnal forcing (similarly, the LT component has a significant amount of annual energy). Model performance metrics for the DU component is misleadingly optimistic because it mostly reflects the 24 hour and annual forcings embedded in both the observations and model values. For periodic processes, metrics derived from the amplitude and phase can be more informative.

**Response.** We have added the comment to the text (see line 332). We reserve to expand to those metrics in future analysis.**

4. The variance of the ID term is very small. Therefore, although the paper shows both the fraction of variance due to each component and the error terms, it should be pointed out that the small errors in the ID component are quite large relative to the total amount of ID variability.

**Response. We have included some comments in the text (see line 344)**

5. Model/observation correlation as a stand-alone metric can be informative as it shows whether the model can reproduce patterns seen in the observations. For example, the ID component, as noted, has small errors, but for individual monitoring sites (not spatially averaged), correlation between modeled and observed ID is often quite low and insignificant (there often appears to be no relationship between the two). On the other hand, correlation tends to improve as time and space scales increase, often leaving the LT component with the best agreement in terms of correlation.

**Response.** We have included the values of the correlation coefficient directly into the error breakdown plots, therefore allowing for a compact view of the error magnitude and associativity value.

**Editorial comments**

There is some confusion in the text when discussing bias. Figure 2 actually shows squared bias, though the discussion seems to be referring to both bias and squared bias.

**Response.** We have clarified the figure 2 by adding 'bias2' in the legend and clarified the discussion in the main text

Line 263: should read "has little impact" or "has negligible impact"

**Response. Done**

Line 283: The statement ending on this line could use a reference.

Response. It is derived based on the analysis of the current study.

Line 305: What is meant by "sparseness of the modeled values"?

**Response.** The sentence has been removed from the text.

Line 452: should have a period at the end

Line 457: should have a period at the end

**Response. Done**

Figure 1. Panels do not have 'a)' and 'b)' labels.

**Response. Done**

Also, if it's not too much trouble, invert the legends so that the colors appear in the same order as they do in the bars.

**Response. Done**

Figures 3 and 4 are very difficult to look at. When error terms are small it is hard to tell where the intersection is. Zooming in would help with better image resolution.

**Response.** We have improved the resolution of the figures and added an zoomed example for clarification. Rather than the individual station's error, we wish to convey the message contained in the method.

Figure 8: Caption should read 'from right to left'.

**Response. Done**

**Editor's comments left open from to quick report**

Figure 3 – 6: please use larger font to show the title of each panel ("MSE of spectral components ..."). It may be sufficient to simply show the model name, AQMEII phase, and continent as title in each panel.

Response. Done

Figure 4, 6: please make sure that the color scale on the right does not overlap the geographical features in the Northeast corner of the map. **Response.** Done

Figure 8: Please add a title and units to the y-axis **Response.** Done

Figures S4 – S7: Please add a title and units to the y-axis **Response.** We have removed the figures for the supplementary material, as they did not add much to the discussion with respect to the ones already presented in the paper

Is Table 2 accidentally split in two sections? (pages 20 and 21) **Response.** The table is split in two parts, each one describing the models participating to AQMEII 1 and AQMEII2, respectively. We have specified it in the tables.